# Assessing Alignment of Patient and Clinician Perspectives on Community Health Resources for Chronic Disease Management

**DOI:** 10.3390/healthcare10102006

**Published:** 2022-10-12

**Authors:** Kameswari A. Potharaju, Jessica D. Fields, Anupama G. Cemballi, Matthew S. Pantell, Riya Desai, Antwi Akom, Aekta Shah, Tessa Cruz, Kim H. Nguyen, Courtney R. Lyles

**Affiliations:** 1School of Public Health, University of California, Berkeley, CA 94704, USA; 2Department of Medicine, Center for Vulnerable Populations at Zuckerberg San Francisco General Hospital, University of California, San Francisco, CA 94143, USA; 3Division of General Internal Medicine, Zuckerberg San Francisco General Hospital, San Francisco, CA 94143, USA; 4Department of Pediatrics, University of California, San Francisco, CA 94143, USA; 5Department of Family and Community Medicine, Center for Health and Community, University of California, San Francisco, CA 94143, USA; 6Social Innovation and Universal Opportunity Lab (SOUL), University of California, San Francisco and San Francisco State University, San Francisco, CA 94132, USA; 7Department of Africana Studies, College of Ethnic Studies, San Francisco State University, San Francisco, CA 94132, USA; 8Streetwyze, Digital Organizing Power Building and Engagement Labs (DOPE), Oakland, CA 94612, USA; 9Department of Epidemiology and Biostatistics, University of California, San Francisco, CA 94143, USA

**Keywords:** chronic disease, social determinants of health, implementation science, qualitative research, electronic health record, community health resources, community health networks, mixed-methods research

## Abstract

Addressing social determinants of health (SDoH) is associated with improved clinical outcomes for patients with chronic diseases in safety-net settings. This qualitative study supplemented by descriptive quantitative analysis investigates the degree of alignment between patient and clinicians’ perceptions of SDoH resources and referrals in clinics within the public healthcare delivery system in San Francisco. We conducted a qualitative analysis of in-depth interviews, patient-led neighborhood tours, and in-person clinic visit observations with 10 patients and 7 primary care clinicians. Using a convergent parallel mixed methodology, we also completed a descriptive quantitative analysis comparing the categories of neighborhood health resources mentioned by patients or community leaders to the resources integrated into the electronic health record. We found that patients held a wealth of knowledge about neighborhood resources relevant to SDoH that were highly localized and specific to their communities. In addition, multiple stakeholders were involved in conducting SDoH screenings and referrals, including clinicians, system navigators such as case workers, and community-based organizations. Yet, the information flow between these stakeholders and patients lacked systematization, and the prioritization of social needs by patients and clinicians was misaligned, as represented by qualitative themes as well as quantitative differences in resource category distribution analysis (*p* < 0.001). Our results shed light upon opportunities for strengthening social care delivery in safety-net healthcare settings by improving patient engagement, clinic workflow, EHR engagement, and resource dissemination.

## 1. Introduction

Individuals’ physical, social, and mental well-beings are greatly shaped by social and environmental conditions such as transportation access, public safety, socioeconomic status, available green space, and housing—which have collectively become known as the social determinants of health (SDoH) [1]. SDoH have a significant impact on the management of chronic diseases and the delivery of health promotion interventions, exemplified by diabetes treatment and management for some of the 28 million community health center patients in the United States [2]. In 2020, 34.2 million people in the United States, representing 10.5% of the population, had diabetes [3]. The prevalence of type 2 diabetes, like other chronic diseases, is disproportionately high among people of color and individuals of low socioeconomic status. Patients who visit community health centers and other safety-net healthcare settings, who are also more likely to be uninsured and have less continuity of care, are nearly twice as likely to have or be at risk for diabetes compared to the overall low-income population [4,5,6,7]. A large proportion of patients served within safety-net settings have complex medical needs, often with multiple chronic diseases and/or behavioral health conditions [5]. Increasing complexity of a patient’s social needs, particularly relating to SDoH such as insurance status and primary language, is associated with poorer health outcomes in primary care settings, emphasizing the importance of addressing SDoH within chronic care management at safety-net clinics [5,8]

Health outcomes for chronic illnesses are thus not just a function of medical care, but also the social determinants that affect care management [6,9,10,11]. Connecting patients to social needs resources, alignment of social and healthcare service organizations, and activities to promote social needs resource development are therefore necessary aspects of improving healthcare delivery for patients [12]. Particularly in under-resourced settings such as safety-net clinics, addressing SDoH for patients with greater social complexities can be challenging for clinicians. While safety-net healthcare settings are uniquely positioned to address social needs based on their expertise, services, and position in the community, they also often lack the resources to provide adequate support to all those who need it.

One area of opportunity for the integration of SDoH into clinical care is through digital health. Digital health is broadly defined as “the field of knowledge and practice associated with the development and use of digital technologies to improve health across the full range of health technologies introduced into care, including telehealth, mobile health apps and wearable technologies, and online health services and tools” [13]. Specific to digital health tools for assessing social needs, the utilization of electronic health records (EHRs) to capture SDoH data is increasingly being used to screen and then refer patients to additional health and social resources within the community [14]. However, SDoH documentation practices widely vary across safety-net clinics, whose approaches to SDoH have thus far been unstandardized and manual [15,16]. There is great variation in how clinicians collect data on SDoH; some EHRs contain structured fields for entering SDoH information (such as ICD-10 codes), while others rely on clinicians entering free-text notes during patient visits. Each of these tools are used to varying degrees by individual clinicians and care teams, particularly due to the differences in clinical workflows within a given health network and existing digital platforms within their setting. Moreover, many clinicians take varying actions based on this SDoH data to improve the quality of patient care. The compilation and organization of SDoH resource lists in EHRs is therefore an opportunity to better understand how clinical clinicians approach SDoH screenings and referrals.

This study addresses the gap in knowledge regarding the current state of implementation and optimization of health technology tools and social resource lists to address SDoH among patients with diabetes in safety-net clinics, as well as the distribution and categorization of SDoH community resources. Understanding what makes communities vulnerable and/or resilient to SDoH is critical to developing mitigation actions that can streamline information flows between healthcare stakeholders as well as prioritize and align needs between patients and clinicians. As such, this study examines the extent of alignment between patient and clinician perspectives of SDoH resources and referrals in clinics within a public, safety-net healthcare delivery system.

## 2. Methods

This project utilized qualitative data, supplemented by quantitative analysis, collected as a part of the first phase of a National Library of Medicine-funded R01 entitled Mapping to Amplify the Vitality of Engaged Neighborhoods (MAVEN). This initiative aims to design an online tool for community members and leaders seeking to access neighborhood health resources to improve chronic disease management.

### 2.1. Study Setting and Sample

This study took place within the San Francisco Health Network (SFHN), the city’s public healthcare delivery system. SFHN serves approximately 80,000 primary care patients throughout the city and county of San Francisco, the vast majority of whom are uninsured or insured via Medicaid. SFHN transitioned to the Epic EHR in August 2019, which overlapped with the time period of this study.

### 2.2. Data Collection

This study used multiple types of data collection procedures with several audiences to provide a multi-faceted examination of health and social resources both within and outside of healthcare settings.

Within SFHN, we first identified patients for the qualitative portion of this study, focusing on patients with a chronic disease with ongoing care within the delivery system. Following provider review of eligible patient lists and brief phone screenings, we identified patients who (1) preferred English as a primary language (given that language accessibility of health resource directories were often not fully robust in other languages); (2) had a diagnosis of either diabetes or prediabetes, verified in the SFHN EHR (as an indicator of routine engagement with primary care); and (3) residency in underserved San Francisco neighborhoods (i.e., Bayview–Hunter’s Point, Mission, Tenderloin, Western Addition, and Visitacion Valley), per the definition used by the San Francisco Planning Department [17]. Primary care clinicians were identified by direct outreach to SFHN clinics in these same neighborhoods. Finally, we also conducted 8 neighborhood tours with community leaders working in community-based organizations that served the same neighborhoods within San Francisco. These community leaders were recruited via an existing network of organizations working on health prevention and promotion throughout San Francisco. More details about the sampling methodology have been previously reported [18].

The UCSF Institutional Review Board approved this study (IRB#18-25696). All participants provided informed written or verbal consent. The qualitative work was guided by the Standards for Reporting Qualitative Research [19].

### 2.3. Qualitative Methods and Procedures

#### 2.3.1. Patient Interviews

Our qualitative approach was informed by grounded theory [20]. With the patient sample, we conducted semi-structured interviews, primary care visit observations, and walking tours of patients’ neighborhoods, which were all audio-recorded. Patient interviews, lasting 90–120 min, were conducted in-person in patients’ homes or public spaces within their neighborhoods in 2019. Clinic visit observations were also in-person for four of the patient participants at the Richard Fine People’s Clinic and Tom Waddell Urban Health Center, also within 2019. Neighborhood tours lasted for 60–120 min.

In the interviews, patients described their experiences living with a chronic disease, what being healthy means to them, and how they access information about health-related resources. Patients also described their community and identified places in their neighborhood that they perceive as contributing or not contributing to their health. During the neighborhood tours, patients and community leaders pointed out frequently visited locations related to everyday life and health management. Lastly, clinic observations included interviews after each visit about the patient’s perception of the visit and their relationship with the healthcare clinician. Members of the study team with previous qualitative research expertise within this community (CRL, KHN, JRF, and AGC) completed the data collection with patients and community leaders.

#### 2.3.2. Clinician Interviews

In addition to the four primary care clinic observations discussed above, we also interviewed 7 clinicians from SFHN about their experience and perspectives on how theyaddress patients’ social needs and utilize community health resources in their everyday practice. One to two members of the study team led individual, semi-structured interviews with each clinician participant either in-person at their healthcare setting or remotely using a web-based conferencing tool (Zoom Video Communications, San Jose, CA, USA) in late 2020 and early 2021.

In clinician interviews, participants were asked about their experiences working with other clinicians in the clinical setting related to addressing patients’ social needs, their knowledge of local community resources for patients’ social needs, and major stakeholders within their clinical practice involved in addressing SDoH. Additionally, clinicians described their process of conducting referrals to community organizations, challenges they face in following up and coordinating with patients, and the use of resource lists and other social referral tools within the EHR. A similar group of study team memberswith qualitative data collection expertise (CRL, KHN, JRF, AGC, and KAP) planned and completed the data collection with clinicians.

#### 2.3.3. Qualitative Analysis

We conducted a thematic analysis of the qualitative data collected, using open coding of ideas and concepts that emerged [21]. Transcripts of all qualitative data—interviews, neighborhood tours, and clinic visit observations—were read multiple times and coded initially by authors KHN, JDF, AGC, RD, and CRL, using an inductive thematic analysis approach [18]. Authors KP and CRL conducted another round of coding, employing an additional codebook based on an inductive coding approach focused on the knowledge and utilization of community resource referrals for the purpose of this study. Based on this thematic analysis, authors KP and CRL developed final themes and selected illustrative quotes and examples. All authors, including clinician insights from MP and additional qualitative and community-based expertise from AA, AS, and TC, provided input on the final codebook and themes in a group-based consensus approach.

### 2.4. Quantitative Methods and Procedures

#### 2.4.1. Mapped Community Resource Data

All mapped community resource data was collected and/or entered into an online platform called Streetwyze [22]. The platform allowed patients, community members, and the research team to share local knowledge relevant to SDoH that are highly localized and specific to communities [23]. Streetwyze’s mobile mapping and data visualization tools enabled tagging of “visible” neighborhood resources, such as community or health centers, as well as more “invisible” neighborhood resources that can easily go unnoticed or be harder to detect, such as parks or art murals [24,25]. More specifically, the platform allowed the research team to map resources by geographic location and to document the type/category of the resource, while simultaneously providing additional contextual data about the resource if relevant (such as narrative review or image related to the resource).

We had two sources of mapped community resources to count and categorize quantitatively, to complement and triangulate our qualitative data collection and analysis;we used the Streetwyze platform for both data sources. First, we examined all mapped community resources that came out of interviews and neighborhood tours with patient and community leaders. We used Streetwyze during the neighborhood tours and entered data prospectively into the platforms, ensuring consistency by having research staff proactively and systematically enter resources. We used the existing categories within the platform to initially tag each resource, and mapped the resource to street address as well as latitude and longitude coordinates.

Second, we extracted from SFHN all community-based resources entered within the existing EHR-based directory. This was a comprehensive list of social and community services within San Francisco, started through a contract with an external vendor HealthLeads [26]. and then edited and added to by frontline clinical staff over time. For these SFHN data, we manually entered all EHR directory -resources into the Streetwyze platform, using the existing categories identified within the EHR and adding geospatial locations.

#### 2.4.2. Analysis

To quantitatively analyze the community resource data, we first completed a final refinement of the resource categories within the combined database. We consolidated the code categories across the two datasets into a final set of resource types, primarily collapsing resources into the overarching category type. One author (KP) generated the first list of resource categories and additional co-authors (CRL, KHN, and AGC) participated in refining the final category definitions. These categories represented the broadest and most common resource categories, such as food and housing resources.

Finally, to examine whether the two datasets of community resources differed, we calculated the proportionate percentages of each category across each data source to compare the distribution of resource types by data source. Given the small cell sizes in some resource categories, we used a Fisher’s exact two-sample, chi-squared test to compare resource categories between the two datasets.

## 3. Results

### 3.1. Participant Characteristics

The study sample consisted of 10 patients and 7 clinicians. Table 1 describes the participants’ characteristics. Patients were majority people of color and female with an average age of 62 years. Most earned less than USD 20,000 annually and had multiple chronic comorbidities. Clinicians were either primary care physicians or nurse practitioners, all of whom practiced within the San Francisco Health Network.

### 3.2. Qualitative Findings

Our interview findings elucidated two major themes encapsulating the experiences of patients and clinicians in navigating social resource linkages in safety-net settings.

#### 3.2.1. Theme #1: Misalignment between the Prioritization of Social Needs by Clinicians and Patients

##### Patient Descriptions of Social Needs Resources

Based on patient interview data, patients’ descriptions of commonly used social needs resources demonstrated localized, neighborhood-specific knowledge relating to the usability and quality of community health resources. For example, Patient 3 described frequenting a particular grocery store in their area that provides fresh produce and accommodates individual requests:

“*…there’s a little market over here. If you want something that they don’t carry, they accommodate you and bring it in. They have some fresh fruits and vegetables a block further down. So, it’s much more convenient here for procuring food.*”—Patient 3

Patient 4 identified several grocery stores in the area that accepted fruit and vegetable vouchers. Patient 7, on the other hand, described avoiding a certain grocery store in their area because it sold low quality, expensive produce. Similarly, Patient 8, who liked to cook, identified a local wellness center that offered healthy cooking classes that allowed them to learn how to incorporate more nutritious foods in their diet. Finally, patients also described benefiting from clinical-social needs interventions that supported their individual needs. Patient 2 reported a community-based organization (CBO) employee who provided food delivery services weekly, while Patient 3 talked about a nurse at their living facility who personally brought them medication on a daily basis in order to facilitate medication adherence:

“*We have a house nurse on premises and I finally confessed to her what I was doing and that I needed help being more religious about taking the medication so she brings it down every day, she’s here five days a week, brings it down every day and I take it while she’s in the room.*”—Patient 3

Overall, the resources mentioned by patients reflected localized knowledge of their community and individual needs. Patients know highly granular details about their neighborhood and the utility of its resources. Because of their daily lived experiences in these communities, patients were able to identify opportunities or barriers to resource utilization that are more personal and qualitative. This high degree of neighborhood specificity affects how patients navigated and utilized community health resources.

##### Clinician Descriptions of Social Needs Resources

In interviews, clinicians described focusing on patients’ social needs such as food and housing that can be addressed using existing longstanding partnerships with community-based organizations or services already offered by the hospital system, as opposed to others, such as education or art resources that would be more difficult to address. Several clinicians referenced a common sentiment of “no screening if there is no intervention,” referencing the practical and ethical concerns of screening for certain social needs without having any resources to actually provide patients to address those needs.

“*‘What are your needs for shelter and food?’… I think those are the priority, but I think what ends up happening is if the patient doesn’t mention it, and you’re not actively looking for it, it doesn’t always come up. Sometimes it will happen that, “I know we have this program. Let’s be on the lookout for patients who need this thing…” It’s not that you’re not screening, but it’s like you’re particularly on the lookout when there’s something to actually offer. I don’t think that’s a conscious way of prioritizing, but I think the reality is that you might be more likely to be aware of it or asking about it when there is some particular resource to be had or to be used.*”—Clinician 3

Due to reasons such as the limited time available in clinic visits, clinicians reported triaging and prioritizing among patients’ often numerous social needs, focusing on what was most conspicuous, urgent, and easily addressed in a short clinic visit. Additionally, this process of prioritization varied from clinician to clinician. For example, Clinician 2 mentioned that she focuses on issues of hygiene and housing as they are often most apparent within a short clinic visit. On the other hand, Clinician 5 said that she only sends referrals to food resource organizations, which are most accessible for her since the process of referring patients to social resources itself is often prohibitively complex and highly variable:

“*Well, the only meal resource that I know how to refer to that I personally actually know how to do is the project in San Francisco, the Project Open Hand, which is just a single form that you fill out and fax in all the information. It’s very straightforward and the eligibility criteria are on the application. The fax number is on the application. Everything is right there. So, I don’t have to think very much about it. I could just go and fill it out.*”—Clinician 5

Moreover, while clinicians broadly approached prioritizing patients’ social needs based on ease of urgency and ease of referral-making, patients tended to have neighborhood-specific knowledge about the utility of both formal and informal local resources. Table 2 provides some examples of how clinicians and patients addressed a similar social need in different ways based on their understanding of social needs resources and approach to prioritizing social needs.

#### 3.2.2. Theme #2: A Myriad of Clinical Workflows for Linking Patients to Social Resources

Most clinicians reported learning of social needs resources based on word of mouth from other clinicians, and there was minimal engagement with EHR tools designed to record social needs screenings. Clinician-clinician communication methods varied significantly across clinics; some utilized text message groups to interact with fellow clinicians and share resources, while others met with their clinician team at least once a day in a “huddle” to informally discuss the needs of the patients being seen that day:

“*Then I will often quickly go around to the different team members and do a mini huddle with them because I’ve missed the big group huddle, but I will say what makes it—we tend to know our patients very well… [I] know most of the people who are coming in and what their needs are… it makes the huddle actually more meaningful because we tend to know everyone*.”—Clinician 2

In addition to interacting with fellow clinicians, clinicians relied on system navigators and community-based organizations with expertise in social needs resources to manage resource linkages. Many of the healthcare clinicians we spoke to did not actually process social needs referrals themselves, instead deferring to the clinic’s social worker, for example. These intermediaries varied from clinic to clinic and included social workers, case managers, patient navigators, and volunteer groups. Social needs screening were done at any point in the visit: during an intake by a medical assistant, in the clinic visit with a physician or nurse practitioner, or after the visit with a system navigator. Clinician 1 described one such student volunteer navigator group:

“*…they have contributed a lot in terms of the number of patients that they interact with, the proactive screening, the refining of social risk screening tools… They still contribute hugely to the resources both in [the EHR] and certainly to the one that you’ll eventually see that is a Google Doc that everybody’s been using.*”—Clinician 1

Given the lack of standardization across clinics even within a single health network, how a patient’s social needs information (including identified needs, referrals, and available resources) was stored and shared among various stakeholders was inconsistent and difficult to access by other clinicians. Clinicians frequently mentioned that there is a lack of a centralized resource system that can be universally accessed by all clinicians within the network. Although the EHR contained a resource list, the interview data showed that there was no standardized utilization of or training on using this resource list; several clinicians interviewed had never heard of it before. Clinician 3 pointed out that a major challenge to such a tool would be having someone updating the lists consistently:

“*A big thing is these things are often self-limited, and they change all the time and don’t have somebody who can update them. One of the things that we’d always thought about are figuring out the exercise resources that are available for the patients with low income, and you’d think that rec and park might do that, but it’s not easy, and they’re not coordinated. We’ve had this dream of having a central resource bank or something. So, yes, it would be great to try to coordinate our efforts.*”—Clinician 3

Some clinicians had been taking steps towards developing resource lists for their own clinics. Clinician 3 developed a resource list for patients in the diabetes clinic, specifically addressing common social needs for patients with diabetes. Similarly, Clinician 5 created a resource list shared via Google Docs for clinicians during the COVID-19 pandemic for a variety of resources. However, both clinicians pointed out that the responsibility for upkeep is diffuse as there is no individual or organization designated to periodically review these resource lists.

Another barrier to social needs resource utilization reported by clinicians was patients’ inability to access and leverage these resources. Patients could face challenges every step of the way, from choosing a resource, reaching out, to receiving the services. Clinician 5 described how large resource lists are often overwhelming to patients, who may not know which resources they are eligible for and would benefit from. Clinician 3 also mentioned the importance of culturally appropriate and accessible resources for patients; many resource lists are not compiled with patients with low-income or from racial/ethnic minority groups in mind. For example, Clinician 7 mentioned a particular challenge for non-English-speaking immigrant patients while accessing medications:

“*I think something that I’ve experienced this past week a lot lately is our patients who are Hispanic or Spanish-speaking patients… healthcare is very different in their countries. When they come here and they need refills on medication, they think that they need to come to the clinic and request it. So oftentimes, they go without medication for two or three or four weeks… there’s a lot of education involved around letting patients know from the very beginning like your clinician has given you a year’s worth of medication at the pharmacy.*”—Clinician 7

Clinicians discussed accounting for cultural competence when identifying and referring ethnically diverse, non-English speaking patients to community resources, as patients may have varying degrees of literacy when navigating the U.S. healthcare system.

Lastly, clinicians struggled to conduct follow-ups to close the loop with patients to determine whether or not their social needs have been met after a social referral. Given the limited capacity of clinic resources and staff, clinicians were rarely able to check-in on patients post-referral to evaluate patient utilization of resources.

“*I know [if they access a resource] when I check in on them, but like there is not a good system set up… the thing that’s tough is you’re exchanging information…that’s identifying healthcare systems with community-based organizations. With HIPAA [the Health Insurance Portability and Accountability Act] it’s very hard to do that in a warm hand off way. So, most of the time no. We give them the info I’d say, I give them the info, or my team does, and we hope for the best.*”—Clinician 6

Given the lack of standardization in SDoH screenings and referrals across clinics, it was challenging for clinicians to identify whether or not a patient had been successfully connected with the chosen resource until the patient returns to the clinic. Moreover, there was little to no direct contact between care clinicians and CBOs in a standardized manner, such as via the EHR, which added to the gap in communication surrounding patient follow-ups. In addition to technological barriers, regulations surrounding private health information were an obstacle to CBOs and clinicians interfacing to ensure that patient needs are being met.

### 3.3. Quantitative Findings

Next, we found differences between healthcare vs. community-oriented resource lists in our descriptive quantitative analysis.In total, there were 441 resources in the SFHN EHR list and 250 resources in the online mapping database from neighborhood tours.

Table 3 contains the number and proportions of resources by category across the two resource lists, summarized into the major categories such as food, health, education, transportation, housing, social services, and other. We found a statistically significant difference between the categorical distribution of the community-generated and health system-generated resources (*p* < 0.001). The majority of the patient/community-generated resources frequently utilized fell into the categories of food, health, and public spaces. Alternatively, the resources in the SFHN EHR system primarily fell into the categories of social services, health, and housing, with no resources representing environmental SDoH or public spaces. Moreover, substance use disorder services and legal services were not explicitly identified in the patient/community-generated resource list. This finding correlates with Theme #1 in our qualitative analysis, where we discovered a misalignment of resource prioritization between patients and clinicians.

## 4. Discussion

Patients within this study demonstrated locally specific knowledge of community health resources, based on their community context and individual needs, that was key to their ability to utilize and benefit from community health resources. This was the case whether the resource was recommended or referred to them from their clinician or healthcare system as well as whether they knew or learned about the resource within their own community. However, clinicians within the study tended to take a different approach to community resource referrals, understanding how to prioritize their limited time with patients to focus on social needs referrals well matched for support from existing health system-CBO partnerships. This misalignment between patients and clinicians was upheld in our examination of the quantitative comparison of resource distribution, where we found a statistically significant difference in the distribution of types of community resources between patient- and clinician- generated lists. Finally, this study identified key obstacles in the clinical workflow that impede the process of community resource referrals within the context of a busy primary care practice in a public healthcare delivery system, such as: the multiple staffing models for screening and referrals; absence of trusted, centralized resource lists; a dearth of updated and easily accessible culturally appropriate resources; and the overburdening of limited clinic staff with additional tasks.

Importantly, this is one of the first papers to our knowledge that compared patient/community and provider perspectives on social needs and community resources, as much of the previous work in this space has focused on one audience at a time [27,28]. The ability to understand perspectives across audiences is critical, given that we are expanding attention to patients’ social needs within healthcare settings while simultaneously breaking down the siloes between medical and public health approaches to community health—particularly as we move beyond the COVID-19 pandemic. Furthermore, this study field highlights the importance of improving patient-clinician communication about SDoH and facilitating the incorporation of patient input into the referral process.

As described by previous research, SDoH interventions in clinical settings—sometimes referred to as “social prescribing”—represent a significant area of opportunity for improving health outcomes for marginalized populations [29]. Our research supports the existing literature identifying the wider spectrum of strategies for improving social prescribing practices, such as standardizing SDoH data collection and community resource linkages, with a particular focus on EHR tools; expanding the healthcare workforce to better accommodate care coordination; and strengthening partnerships with CBOs and other stakeholders in the clinic and community [16,30,31,32]. Successful interventions and existing tools/platforms for community resource referrals have addressed similar themes, but often with a focus on a singular aspect such as social needs screening or utilizing patient navigators, rather than the holistic perspective that is emphasized within this study. The clinicians in this study also described a need for a SDoH screening and referrals to go further and optimize health resources best suited to patients based on location, lived experiences, and cultural concordance, alongside usability within real-world practice. In addition, this study furthers the field by emphasizing patient priorities, such as the importance of localized neighborhood knowledge tied to patients’ lived experiences that should be integrated into clinical care in order to better serve patients.

Moving forward, while online social resource platforms continue to spread, additional research is needed in this space, particularly on the specific workflows, staffing models, and reimbursement structures for social prescribing and community resource referrals within safety-net clinics which have a medically and socially complex patient population. More work is also needed to determine ideal processes for data exchange between healthcare systems and the CBOs that address mutual patient/client social needs to ensure information is up to date and to reduce burden on both clinicians and CBO staff facilitating referrals. In addition, the job descriptions and division of responsibilities among social workers, patient navigators, and other intermediaries need to be examined for sustainability and scalability of best practices to ultimately identify the correct stakeholders to lead SDoH screening and referral processes. Seeking and incorporating the perspectives and feedback of these interprofessional clinicians could greatly improve our understanding of optimal clinical practices for addressing SDoH.

This study has several limitations, largely with regard to generalizability. The perspectives shared in this paper represent patients and clinicians in a single health system in one major U.S. city and therefore may not be representative of safety-net settings in other regions due to the unique demographic and geographic characteristics of San Francisco. Additionally, the study sample was limited to a small number of English-speaking patients and clinicians within this setting, excluding the perspectives of patients with limited English proficiency and additional stakeholders such as social workers and CBO staff who often interact directly with patients by providing social needs resources. In addition, we did not explore deeper dimensions of patient identities, such as gender identity or race/ethnicity. This study’s strengths, however, include: the diverse racial/ethnic and socioeconomic characteristics of the participant population; the use of qualitative and quantitative data derived from multiple sources (interviews, neighborhood tours, physician visits) that offer a more nuanced understanding of participant experiences; and the triangulation in both the methodology and diverse team of investigators to represent a broader range of perspectives and reflexivity.

Ultimately, there is a significant need and opportunity to address SDoH for patients in safety-net settings. In the process of implementing digital tools and improving clinical workflows to link patients to community health resources, it is essential to prioritize multi-stakeholder involvement and input from patients, community leaders and organizations, clinicians, and other clinic staff. Healthcare institutions must take initiative to prioritize partnerships and interventions to improve the health outcomes of patients from marginalized communities.

## Figures and Tables

**Table 1 healthcare-10-02006-t001:** Participant Characteristics.

Patients (*n* = 10)						
**Patient #**	Race/Ethnicity	Gender	Age Group	Income	Neighborhood	Chronic Diseases
1	Black/African American	Female	60–64	Not answered	Bayview–Hunter’s Point	Diabetes
2	Black/African American	Male	60–64	Not answered	Bayview–Hunter’s Point	Diabetes, heart disease, high blood pressure, heart failure, asthma/COPD, chronic pain in back/legs
3	White	Female	65–69	Less than USD 20,000	Tenderloin	Pre-diabetes, heart disease, high blood pressure, asthma/COPD, depression, anxiety
4	Asian or Pacific Islander	Female	60–64	USD 20,000–40,000	Tenderloin	Diabetes, heart disease, high blood pressure, asthma/COPD, depression, anxiety, osteoarthritis, PTSD
5	Black/African American	Female	70–74	Less than USD 20,000	Western Addition	Diabetes, high blood pressure, neuropathy
6	Black/African American	Male	55–59	Less than USD 20,000	Tenderloin	Diabetes, high blood pressure, depression, anxiety, cataracts
7	Black/African American; Multi-Ethnic	Female	60–64	Less than USD 20,000	Bayview–Hunter’s Point	High blood pressure, depression, anxiety, rheumatoid arthritis, fibromyalgia, lupus
8	American Indian/Native American	Male	60–64	USD 20,000–40,000	Excelsior	Diabetes, high blood pressure, asthma/COPD, chronic kidney disease, depression, high cholesterol, liver disease, nasal inflammation
9	Black/African American	Female	55–59	Less than USD 20,000	Bayview–Hunter’s Point and Lakeshore	Diabetes, high blood pressure, chronic kidney disease, high cholesterol
10	Hispanic/Latinx	Female	45–59	Less than USD 20,000	Tenderloin	Pre-diabetes, depression, anxiety
**Clinicians (*n* = 7)**						
**Clinician #**	**Role**			**Role Description**		
1	Primary care physician			Family medicine physician		
2	Primary care physician			Internist at safety-net clinic		
3	Nurse practitioner			Family practice nurse practitioner at primary care center		
4	Nurse practitioner			Family nurse practitioner at community health center		
5	Primary care physician			Resident physician in family medicine program		
6	Primary care physician			Pediatrician		
7	Nurse educator			Nurse educator in a public healthcare delivery system		

**Table 2 healthcare-10-02006-t002:** Comparisons of Provider and Patient Perspectives on Social Needs.

Social Need	Clinician Perspective	Patient Perspective	Misalignment in Perspectives
Food	Clinician 2 reported that if they hear a patient is struggling to access fresh fruits and vegetables, they will refer the patient to a social worker who could arrange for the patients to receive groceries through the clinic’s food pharmacy.	Patient 7 reported that a particular location of a chain grocery store has substandard or rotten meat and vegetables. Patient 4 mentioned a farmer’s market held twice a week in their neighborhood.	While clinicians seek to connect patients with a readily available food resource, patients are concerned with the quality and proximity of food resources.
Physical Activity Spaces	Clinician 4 described making referrals to an exercise coaching program at the hospital.	Patient 6 described visiting a local community center for tai chi classes. Patient 10 mentioned attending free yoga and holistic healing classes in their neighborhood.	While clinicians sought to connect patients with established exercise programs through the health network, patients described attending exercise classes that were located in their neighborhood and offered more options for activities.
Transportation	Clinicians 3 and 4 discussed referring patients to the behavioral health team for resources such as transportation vouchers.	Patient 1 described using certain bus lines to visit the hospital. Patient 9 discussed the challenges of obtaining an electric wheelchair in order to utilize transportation.	While clinicians sought to provide financial support for transportation, patients expressed concern about the usability of transportation options, beyond affordability.
Resource Lists	Clinician 5 described creating a resource list on Google Docs based on suggestions from fellow clinicians and outreach from CBOs for clinicians to reference.	Patient 5 identified 211 * as a social services resource list available via phone.	While clinicians sought to compile internal lists to be shared with patients on an individual basis, patients described publicly available resource lists. Moreover, patients expressed the importance of word-of-mouth referrals through informal social networks for increasing resource awareness.
Pharmacy Access	Clinician 6 discussed making referrals to certain pharmacies based on what insurance plans they accepted.	Patient 2 reported a preference for a specific pharmacy that organized and packaged pills in a convenient manner.	While clinicians sought to prioritize affordability and accessibility of pharmacy-dispensed medication, patients expressed additional interest in convenient and usable medication packaging.
Community Cohesion	Clinician 2 described the activities led by the Wellness Center at the hospital to foster community togetherness.	Patient 7 discussed experiencing community cohesiveness through an informal “buddy system” to ensure safety for all the seniors in their neighborhood complex.	While clinicians described health network-based opportunities for community development, patients pointed out informal and local resources for improving community cohesion.

* 211 is a hotline for county information and referral services in San Francisco.

**Table 3 healthcare-10-02006-t003:** Analysis of Community-Generated and Health System-Maintained Resource Data.

Category	# of Resources in Community-Generated Data (Percentage) *	# of Resources in Health System-Maintained Data (Percentage)
Food	43 (22.51%)	27 (5.96%)
Health	38 (19.89%)	64 (14.12%)
Education	13 (6.80%)	27 (5.96%)
Transportation	12 (6.28%)	5 (1.10%)
Housing	8 (4.19%)	47 (10.37%)
Social Services	30 (15.70%)	138 (30.46%)
Art	9 (4.71%)	3 (0.66%)
Other (Substance Use Disorder Services, Legal Services, or Environment/Public Spaces)	38 (19.89%)	86 (18.98%)

* While community-generated data included both health-promoting and health-limiting SDoH resources, health system-maintained data included only health-promoting resources.

## Data Availability

The data presented in this study may be available on request from the corresponding author. The data are not publicly available due to containing protected health information.

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
