# Peer review of "Assessing Alignment of Patient and Clinician Perspectives on Community Health Resources for Chronic Disease Management"

_healthcare, 2022, doi:10.3390/healthcare10102006_

Round 1
Reviewer 1 Report
Thank you for the opportunity to review this manuscript. Overall, the paper is well-written, and the complex methodology is explained thoroughly.
My main concerns with this paper are in respect to its generalizability which is discussed satisfactorily in the discussion.
The discussion seems to slant strongly towards the health system perspective, however Theme #1 - the misalignment between prioritization of social needs - is some the most compelling data presented in the study in my opinion. This should be discussed in further detail and future directions for research in this area should be explored.
Furthermore, I am not sure what the exploratory geospatial analysis adds to this work - it is in the title of the paper, but is not commented on in the discussion which focuses primarily on workflow and implementation considerations. I would recommend to either discuss the impact or takeways from these results more explicitly, or remove this information all-together.
Related to the content of this work is a resource that may be useful to the investigators and/or readers with similar experiences - findhelp.org (formerly AuntBetha.org) is a free/open-access resource for locating local services to assist with community needs. This is a user-friendly interface that can be utilized by providers/clinic staff, as well as patients - it contains regularly updated contact and eligibility information, and it also allows for simple documentation of referrals, as well as the ability to "favorite" or take notes for different local resources. This may be something that the authors want to share in the discussion section as a recommendation for readers to pursue - or it may be something that could be implemented in this healthsystem to alleviate some of the concerns collected in this study.
Generally, I have no concerns about the validity of this work, and I think this paper would be of interest to the journal's readership. However, the specificity of the work may not be impactful for other populations.
Reviewer 2 Report
The manuscript “Assessing Alignment of Patient and Clinician Perspectives on Community Health Resources for Chronic Disease Management: A Qualitative Study with Exploratory Geospatial Analysis” is significant in the context of chronic disease management by triangulating both patients' and clinicians' experiences.
Title
In the abstract the authors mention a mixed method, however, in title it mentions a qualitative study. ‘This exploratory mixed-methods study investigates the degree of alignment between patient and clinicians’ perceptions of…’
In methods it is also mention that ‘This project utilized both qualitative and quantitative geospatial data collected’
Please maintain consistency. The author may rephrase the ‘A Qualitative Study with Exploratory Geospatial Analysis’ the above section in a better way.
Abstract
Mention which types of mixed-method approach – Exploratory/Convergent Parallel
Provide some quantitative numbers for your quantitative findings
Introduction
Mention your study objective in the Introduction section.
“Therefore, this study investigated the degree of alignment between patient and clinicians’ perceptions of SDoH resources and referrals in clinics within the public healthcare delivery system.”
Methods
Mention which types of mixed-method approach – Exploratory/Convergent Parallel; justify why you follow such methods.
In table Table 1. Participant Characteristics.
Instead of Age Group, write exact age in years. Change the heading from ‘Gender’ to Sex.
If possible Chronic Diseases heading mention duration. For example
Heading: Chronic Diseases (duration in years); in content: Diabetes (4), heart disease (1), high blood pressure (3), heart failure (0.5), asthma/COPD (7), chronic pain in back/legs (14)
In clinician table mention there sex, age and years of experience.
Mention reporting guidelines in the qualitative study – COREQ or SRQR.
In the analysis mention thematic analysis, and remove “descriptive qualitative analysis".
Results
Table 2. Comparisons of Provider and Patient Perspectives on Social Needs is interesting.
Instead of patient-specific data, summarize the key finding in bullet points under each heading.
Discussion
Integrate the qualitative and quantitative findings.
Is there any gender difference in chronic disease management?
Mention the strength of the study: source, methodological and investigator triangulation; authors' diverse backgrounds. Authors reflexivity.
